# Targeted Maximum Likelihood Learning:
# An Optimization Perspective

**Diyang Li**
Cornell University
diyang01@cs.cornell.edu

**Kyra Gan**
Cornell University
kyragan@cornell.edu

## Abstract

*Targeted maximum likelihood estimation* (TMLE) is a widely used debiasing algorithm for plug-in estimation. While its statistical guarantees, such as double robustness and asymptotic efficiency, are well-studied, the convergence properties of TMLE as an iterative optimization scheme have remained underexplored. To bridge this gap, we study TMLE's iterative updates through an *optimization-theoretic* lens, establishing *global convergence* under standard assumptions and regularity conditions. We begin by providing the first complete characterization of *different stopping criteria and their relationship to convergence* in TMLE. Next, we provide *geometric insights*. We show that each submodel induces a smooth, non-self-intersecting path (homotopy) through the probability simplex. We then analyze the solution space of the *estimating equation* and loss landscape. We show that all valid solutions form a submanifold of the statistical model, with the difference in dimension (i.e., codimension) exactly matching the dimension of the target parameter. Building on these geometric insights, we deliver the *first strict proof* of TMLE's convergence *from an optimization viewpoint*, as well as explicit sufficient criteria under which TMLE terminates in a single update. As a by-product, we discover an unidentified *overshooting* phenomenon wherein the algorithm can surpass feasible roots to the *estimating equation* along a homotopy path, highlighting a promising avenue for designing enhanced debias algorithms.

## 1 Introduction

Plug-in estimation, the approach of first estimating the data-generating distribution and then evaluating the target parameter on this estimate, is a natural strategy for estimating quantities such as quantiles, variance, *average treatment effects* (ATEs), and feature importance measures [1, 2]. Despite widespread use, these estimators often suffer from biases arising from the nonlinearity of parameter mappings, the finite-sample variability inherent to empirical distributions, or model misspecification, which can lead to unreliable inference in high-dimensional or complex settings [3, 4].

To address these limitations, Targeted Maximum Likelihood Estimation (TMLE) [5] offers a principled framework for constructing data-adaptive estimators by combining machine-learning (ML)-based initial estimates with targeted bias correction. At each step, TMLE updates the current distribution estimate along a fluctuation direction to reduce bias for the target parameter, achieving the desired double robustness and asymptotic efficiency under mild regularity conditions [6, 7]. These strengths have driven its adoption across areas including semi-supervised learning [8, 9], personalized medicine [10, 11], algorithmic fairness [12, 13], and off-policy evaluation [14], equipping practitioners with estimators that are both flexible and theoretically grounded.

While TMLE's asymptotic properties (e.g., double robustness and semi-parametric efficiency) are well-established [16], its *finite-sample behavior as an optimization procedure* remains underexplored. Current theory often treats iterative updates as implementation details for solving *estimating*

39th Conference on Neural Information Processing Systems (NeurIPS 2025).

*equations* [16, 17]. While asymptotic guarantees depend on algorithmic convergence, this convergence is typically assumed, except in a few cases with known one- or two-step convergence (e.g., [42, 44]). However, most target parameters lack such one-step guarantees, making algorithmic convergence not just an implementation concern but a fundamental determinant of TMLE's debiasing performance. This gap is especially critical in finite-sample settings like healthcare [18, 19], where asymptotic guarantees offer no guidance for choosing convergence criteria, iteration limits, and tuning parameters. Without optimization-theoretic foundations, implementations remain ad-hoc [20, 21], risking reproducibility and silently degrading estimator quality.

We address this by analyzing TMLE's iterative trajectory and convergence through an *optimization-theoretic lens*. Unlike asymptotic efficiency results, our framework provides actionable insights for practical implementation. Conceptually, the convergence analysis of TMLE shares high-level similarities with that of *expectation-maximization* (EM) algorithms [22], both involving iterative schemes designed to optimize complex objective functions via tractable subproblems using alternating optimization frameworks. However, TMLE's *influence-function* (IF)-driven fluctuations within probability simplex-embedded homotopies pose unique challenges absent in classical EM literature [23, 24]. These technical differences necessitate tailored convergence analyses that differ significantly from those in EM. While recent works have explored optimization aspects for specific problems, e.g., causal effect estimation in exponential families [25] and off-policy evaluation with regularization [14], their scope remains limited to these special settings and does not apply to the general template.

**Our contributions** In this paper, we address a longstanding gap in the theoretical understanding of TMLE by recasting it as an explicit iterative optimization scheme. Our contributions are fourfold:

- **Geometric Insights.** We establish formal connections between different stopping criteria and the resulting algorithm convergence behavior (Theorem 1). Next, we show that each submodel induces a smooth homotopy mapping (or path), embedding each fluctuation into the probability simplex without self-intersection (Theorem 2), precluding cyclic or oscillatory behavior in TMLE. We demonstrate that the set of distributions satisfying the estimating-equation forms a smooth submanifold of the statistical model, whose codimension coincides exactly with the dimensionality of the target parameter (Theorem 3). This structural perspective reveals that TMLE iterates traverse a low-dimensional manifold within the ambient probability space, explaining their effectiveness in navigating a complex landscape with irregular likelihood surfaces (Theorem 3).

- **Convergence Guarantee.** We provide *the first rigorous proof* of TMLE's *global convergence* under mild regularity conditions (Theorem 4). Although this result is asymptotic, requiring an infinite number of iterations, it confirms long-standing empirical observations of TMLE's convergence behavior. Further, we derive explicit sufficient conditions under which TMLE terminates after a single update (Theorem 5). These conditions, based on the initial estimator and IF structure, offer verifiable criteria for simplified TMLE implementations.

- **Overshoot Behavior.** Our analysis reveals an unrecognized overshooting phenomenon, where TMLE may bypass feasible roots along the homotopy path (Theorem 6), potentially affecting finite-sample performance. This insight motivates safeguard strategies such as step-size control or root tracking that retain asymptotic guarantees while improving runtime and stability.

- **Interdisciplinary Impact.** By casting TMLE as an optimization procedure, we bridge statistical estimation with modern optimization theory. Conventional analyses of parametric optimizer using *convexity* typically require the submodel objective to be strictly (or strongly) convex. In contrast, our analyses based on non-intersection do not require such assumptions and thus hold under broader settings. This perspective enables reinterpretations of influence functions via tools like mirror descent, and suggests integrating adaptive acceleration or regularization paths into TMLE, potentially leading to new, theoretically grounded algorithms.

## 2 Preliminaries on Plug-in Estimation

This section introduces notation, reviews plug-in estimation, plug-in bias, and the influence function. Readers familiar with these concepts may skip ahead. Given the dataset $\{O_1, \ldots, O_n\}$ consists of $n$ independent and identically distributed (i.i.d.) observations of a random variable $o$ (e.g., an experimental unit) that fits an *unknown true* distribution $P_0$ with sample space $\mathcal{O}$. Our goal is to *efficiently* estimate a $d$-dimensional target parameter representing some statistical feature of interest (e.g., population mean and average treatment effect).

**Notation** Let $P$ be a probability distribution with density $p$. Abusing the notation, *we use $P$ and $p$ interchangeably to denote the probability measure.* We let $\mathbb{P}_n$ denote the empirical measure (Definition A.5), and $\mathbb{P}_n f := \frac{1}{n} \sum_{i=1}^{n} f(O_i)$. Let $L_0^2(P)$ denote the *space of mean-zero, finite-variance functions* with respect to the distribution $P$, i.e., $L_0^2(P) := \{h : \mathcal{O} \to \mathbb{R} : \mathbb{E}_P h(o)^2 < \infty, \mathbb{E}_P h(o) = 0\}$, where $o$ is a generic random variable drawn from distribution $P$. We use $a \lesssim b$ to denote that there exists a constant $C$ such that $a \leq Cb$. Let $\mathcal{D}_f$ be the Fréchet derivative, and we abbreviate $p_n(o)$ as $p_n$ when no ambiguity arises. We use $\mathbb{C}^j$ to denote the class of mappings that are $j$-times continuously differentiable. A complete notation table is included in Appendix A.

**Plug-in estimation** We consider nonparametric estimation,[1] where the model class $\mathcal{M}$ contains all candidate distributions for $P_0$ on a $\sigma$-finite measurable space $(\Omega, \mathcal{F}, \nu)$, where each element $P \in \mathcal{M}$ admits a Radon-Nikodym density $p = dP/d\nu$ with respect to a dominating measure $\nu$, satisfying $p \geq 0$ $\nu$-a.e. and $\int_{\Omega} p \, d\nu = 1$. *To ease the derivation, we work with the density $p$ of $P$ with respect to a fixed dominating measure. The equivalent definitions in this section can be stated directly in terms of $P$, which is more common in the literature.* We assume $p$ is uniformly bounded:[2]

**Assumption 1** (Uniform Boundedness). *There exists a $C_\infty < \infty$ s.t. $\|p\|_{L^\infty} \leq C_\infty$, $\forall p \in \mathcal{M}$.*

The target parameter functional $\Psi : \mathcal{M} \to \mathbb{R}^d$ then maps each candidate distribution to a corresponding feature of interest (e.g., for $d = 1$, $p \mapsto \mathbb{E}_P[o]$). The *plug-in estimator* $\hat{\psi}_n := \Psi(\widehat{p}_n)$ is obtained by applying $\Psi$ to an empirical estimate $\hat{p}_n \in \mathcal{M}$ of the unknown $p_0$. Let $\psi_0 = \Psi(p_0)$ be the true value of our target parameter.

**Plug-in Bias and Influence Function** While plug-in estimators are often consistent under regularity conditions, they typically fail to achieve *asymptotic linearity* (Definition A.6)[3] due to bias inherited from the initial estimate $\hat{p}_n$[4] and the potential nonlinearity of $\Psi$. Even if $\hat{p}_n$ is consistent at the parametric rate, when $\Psi$ is nonlinear in $p$, estimation errors in $\hat{p}_n$ can propagate nonlinearly through $\Psi$, introducing bias that invalidates a $\sqrt{n}$-linear expansion. As a result, plug-in estimators typically require bias correction to attain asymptotic linearity and efficiency.

A key tool for formalizing this limitation is the *influence function*, which quantifies how sensitive the target parameter $\Psi$ is to small perturbations of the underlying distribution $P$. When $\mathcal{M}$ is nonparametric, the IF of the target parameter $\Psi$ at a distribution $P$, $D_\Psi^*(p)(\cdot) : \mathcal{O} \to \mathbb{R} \in L_0^2(P)$ is unique. We formally introduce the influence function in Appendix A.4, Definition A.9.

To establish the asymptotic behavior of $\hat{\psi}_n$ can then be analyzed through a von Mises expansion. Let $\Psi$ be *pathwise differentiable* (Definition A.8), and consider the perturbation path defined in Eq. (A.14) with $P = \widehat{P}_n$ and $Q = P_0$. Let $P_0$ be absolutely continuous with respect to $\widehat{P}_n$, and $dP_0/d\widehat{P}_n \in L_0^2(\widehat{P}_n)$. The estimation error in $\hat{\psi}_n$ can be decomposed into the following [26, 27]:

$$\hat{\psi}_n - \psi_0 = \mathbb{P}_n D_\Psi^*(p_0) \underbrace{- \mathbb{P}_n D_\Psi^*(\hat{p}_n)}_{\text{plug-in bias}} + \underbrace{(\mathbb{P}_n - P_0)[D_\Psi^*(\hat{p}_n) - D_\Psi^*(p_0)]}_{\text{empirical process term}} + \underbrace{R_2(\hat{p}_n, p_0)}_{\text{second-order remainder}} , \quad (1)$$

where $R_2(\hat{p}_n, p_0)$ is the second-order remainder in the difference between $\hat{p}_n$ and $p_0$. The expansion in Eq. (1) resembles Definition A.6. While standard regularity conditions (e.g., Donsker class requirements and rate constraints on $\|\widehat{P}_n - P_0\|$) suffice to ensure the empirical process term and the second-order remainder are $o_{p_0}(1/\sqrt{n})$,[5] the first-order plug-in bias term typically remains non-negligible without correction.

---

[1]We analyze `TMLE` convergence in nonparametric models, revealing how target smoothness interacts with estimator complexity. Results extend to semi-parametric cases via subspace projections.

[2]This is commonly assumed in prior works to obtain statistical guarantees [5].

[3]Asymptotic linearity ensures $\sqrt{n}$-rate convergence to a normal distribution, with an asymptotic variance determined by the influence function, thereby facilitating efficient estimation and valid statistical inference.

[4]For example, $\hat{p}_n$ may not be consistent at the parametric rate, especially when estimated via neural networks.

[5]We refer readers to Van Der Laan and Rubin [5] and Cho et al. [27] for detailed assumptions.

## 3  Warm-up: TMLE Template

To correct the plug-in bias term, TMLE finds the solution $p_n^*$ that solves the score equation

$$\mathbb{P}_n D_\Psi^*(p_n^*) = \frac{1}{n} \sum_{i=1}^n D_\Psi^*(p_n^*)(O_i) = \int_{\mathcal{O}} D_\Psi^*(p_n^*)(o) p_n d\nu(o) = \mathbf{0}, \tag{2}$$

by iterative refining the initial estimate $\hat{p}_n$.[6] We express this via sample space integration to connect with optimization-theoretic analysis.

Abusing the notation, let $p_n^k$ be the $k$-th iteration of TMLE. At a high-level, after obtaining a "sufficiently good" initial $\hat{p}_n$ (e.g., via ML), TMLE selects a parametric submodel $\{p(\epsilon)\}_{\epsilon \in \mathbb{R}} \subset \mathcal{M}$ (Definition 2)[7] guided by the IF to maximize sensitivity to the target parameter $\Psi$ at $\hat{p}_n$. It then updates $\hat{p}_n$ by fitting $\epsilon$ along this submodel (typically via minimizing a loss function) to obtain an updated estimate $p_n^1$. This procedure is repeated until no further improvement (i.e., no nonzero $\epsilon$) can be found (if achieved). A pseudo-code of TMLE is provided in Algorithm 1. If TMLE terminates after $k$ iterations, the final estimate $P_n^k$ achieves asymptotic efficiency under standard regularity conditions. However, the actual convergence behavior of this iterative procedure remains unestablished.

**Definition 1** (TMLE Loss). *Define the TMLE loss function* $\mathbf{L} : \mathcal{M} \to (\mathcal{O} \to \mathbb{R}_{\geq 0})$ *such that*

$$p_0 \in \arg\min_p \int_\Omega \mathbf{L}(p) p_0(o) d\nu(o). \tag{3}$$

Similar to classical maximum likelihood estimation, we use a loss function $\mathbf{L}(\cdot)$ such that the mapping (3) is minimized at the true density $p_0$. E.g., one may use the negative log-likelihood, $\mathbf{L}(p) := -\log p$.

**Definition 2** (Fluctuation Submodel). *Let $\mathcal{R}$ denote an open subset of $\mathbb{R}^d$. We define a family of fluctuation submodel $\{p(\epsilon) : \epsilon \in \mathcal{R}\}$ (a.k.a. parametric working model) that follows (i) $\{p(\epsilon) : \epsilon\} \subset \mathcal{M}$; (ii) the submodel through $p$ at $\epsilon = \mathbf{0}$; (iii) a linear combination of the components of "score" $d\mathbf{L}(p(\epsilon))/d\epsilon$ at $\epsilon = \mathbf{0}$ recovers the IF (cf. Definition 3).*

$\mathcal{R}$ denotes the set of $\epsilon$ values for which $p_n^k(\epsilon)$ is a proper density. Common parametric submodels include linear (Example 1) and exponential (Example 2).

**Example 1** (Linear Reparameterization). *For $\epsilon \in \mathcal{R}$, the instances of additive perturbation include*

$$p_n^k(\epsilon) \triangleq \left(1 + \epsilon^\top D_\Psi^*(p_n^k)\right) p_n^k. \tag{4}$$

**Example 2** (Exponential Family). *For $\epsilon \in \mathcal{R}$, the instances of exponential tilting include*

$$p_n^k(\epsilon) \triangleq C(\epsilon, p_n^k) \exp\left(\epsilon^\top D_\Psi^*(p_n^k)\right) p_n^k, \tag{5}$$

$$p_n^k(\epsilon) \triangleq C'(\epsilon, p_n^k) \left\{1 + \exp\left(-2\epsilon^\top D_\Psi^*(p_n^k)\right)\right\}^{-1} p_n^k, \tag{6}$$

*for $C(\epsilon, p_n^k)$, $C'(\epsilon, p_n^k)$ be normalizing constants (defined in Definition B.10).*

**Definition 3** (Relaxed Score Condition). *Let $A$ be a constant matrix with $\|A\| < \infty$. TMLE requires that every parametric submodel has a sufficient statistic equal to the IF, i.e.,*

$$\left.\frac{d\mathbf{L}(p(\epsilon))(o)}{d\epsilon}\right|_{\epsilon=\mathbf{0}} \equiv A \cdot D_\Psi^*(p)(o) \quad \text{for all possible values } o \in \mathcal{O}. \tag{7}$$

Definition 3 imposes the statistical connections between the loss and the IF in TMLE. Additionally, we make the following mild regularity assumption, ensuring that the solution of Algorithm 1 is achieved in the interior of $\mathcal{M}$:

**Assumption 2** (van der Laan et al. [6]). *Let $\mathcal{M}$ be a sufficiently rich model class such that the iterated model is locally saturated. Under this condition, the minimization invoked in line 3 of Algorithm 1 admits its solution strictly within the interior of $\mathcal{M}$.*

---

[6]In Eq. (2), we adopt an equivalent integral representation to facilitate the convergence analysis.

[7]We follow the common TMLE convention of using $p$ for both submodels and probabilities, where no ambiguity arises.

---

**Algorithm 1** Targeted Maximum Likelihood Estimator (TMLE)

---

**Input:** Data $\{O_i\}_{i=1}^n$, canonical gradient $D_\Psi^*(p)$ of the interested functional $\Psi$, initial estimator $p_n^0$.

1: $k \leftarrow 0$, initialize $\epsilon_n^0 \neq \mathbf{0}$.
2: **while** $\epsilon_n^k \neq \mathbf{0}$ **do**
3: $\quad \epsilon_n^k \leftarrow \arg\min_{\{\epsilon: p_n^k(\epsilon) \in \mathcal{M}\}} \int_{\mathcal{O}} \mathbf{L}(p_n^k(\epsilon))(o) p_n \, d\nu(o)$
4: $\quad p_n^{k+1} \leftarrow p_n^k(\epsilon_n^k), \ k \leftarrow k+1 \quad \backslash\backslash$ `iterative debiasing`
5: **end while**
6: $p_n^* \leftarrow p_n^{k-1}, \hat{\psi}_n \leftarrow \Psi(p_n^*) \quad \backslash\backslash$ `plug-in estimation`

**Output:** Targeted estimator $\hat{\psi}_n$.

---

## 4 Main Results

This section presents our main theoretical results, framing `TMLE` through an optimization lens for a pathwise differentiable target parameter (Definition A.8). Throughout, we assume that the standard regularity conditions (Assumptions 1, 2) as well as our mild assumptions (Assumptions 3, 4, 5) hold globally unless explicitly stated otherwise.

> **Assumption 3** (Smooth Link). *Let the link function $\check{f}: L_0(\nu) \mapsto L_0(\nu)$ be $\mathbb{C}^2$ smooth. We assume that $p_n^k(\epsilon)$ admits the representation $p_n^k(\epsilon) \triangleq \check{f}(\epsilon^\top D_\Psi^*(p_n^k))$ where $\check{f}(\cdot)$ is injective on its effective domain, namely the linear span of the components of $D_\Psi^*$.*

Assumption 3 ensures that the perturbation parameter $\epsilon$ and the IF $D_\Psi^*(\cdot)$ always appear jointly in the form $\epsilon^\top D_\Psi^*(\cdot)$. Meanwhile, the injectivity imposed on the mapping $\check{f}$ is typically mild in practice and readily satisfied by virtually all mainstream submodels such as those defined in Examples 1 and 2. A formal justification of this injectivity condition is provided in Appendix D.1.

> **Assumption 4** (Differentiability and Lipschitz-in-path). *The mappings $\epsilon \mapsto p(\epsilon)$ and $\epsilon \mapsto \mathbf{L}(\epsilon)$ are of class $\mathbb{C}^2$ and $\mathbb{C}^3$ in $\epsilon$, respectively. The mappings $p \mapsto \mathbf{L}(p)$ and $p \mapsto D_\Psi^*(p)$ are twice continuously Fréchet-differentiable. And for $k \in \mathbb{Z}^+$, $\epsilon_1, \epsilon_2 \in \mathcal{R}$, and $p_1, p_2 \in \mathcal{M}$ we have*
>
> $$\left\| \mathbf{L}(p_n^k(\epsilon_1)) - \mathbf{L}(p_n^k(\epsilon_2)) \right\|_{L^2(\nu)} \lesssim \|\epsilon_1 - \epsilon_2\|_2 ,$$
> $$\left\| \mathbf{L}(p_1) - \mathbf{L}(p_2) \right\|_{L^2(\nu)} \lesssim \|p_1 - p_2\|_{L^2(\nu)} , \tag{8}$$
> $$\left\| D_\Psi^*(p_n^k(\epsilon_1)) - D_\Psi^*(p_n^k(\epsilon_2)) \right\|_{L^2(\nu)} \lesssim \|\epsilon_1 - \epsilon_2\|_2 .$$

> **Assumption 5** (Metric-subregularity of Gradient). *There exists a neighborhood of origin s.t. for $k \in \mathbb{Z}^+$ and all $\epsilon_1, \epsilon_2$ in that neighborhood, with $\nabla_\epsilon \mathbf{L}(\cdot)$ evaluated at $o \leftarrow O_i$, we have*
>
> $$\|\epsilon_1 - \epsilon_2\|_2 \lesssim \left\| \nabla_\epsilon \mathbf{L}(p_n^k(\epsilon_1)) - \nabla_\epsilon \mathbf{L}(p_n^k(\epsilon_2)) \right\|_2 \lesssim \|\epsilon_1 - \epsilon_2\|_2 . \tag{9}$$

Assumption 4 is a standard and widely adopted assumption within the optimization literature. Likewise, Assumption 5 is notably mild within the context of `TMLE` optimization. E.g., one can trivially verify that the log-likelihood under an exponential tilt naturally satisfies (9). Further, the second inequality in (9) is essentially inherited from the vanilla `TMLE` literature (albeit expressed differently).

### 4.1 What do we really mean when we talk about 'convergence'?

In `TMLE` practice, we observe that iterative stopping ("convergence") criteria are often applied inconsistently. To maintain theoretical rigor, we explicitly define `TMLE` convergence as the convergence in probability of the iterates to a deterministic limiting distribution in $\mathcal{M}$. While previous studies like [5, 6] have shown that the convergence implies solving (2), we generalize this result to Theorem 1.

> **Theorem 1** (Stopping Condition Equivalence). *The condition $\lim_{k\to\infty} \epsilon_n^k = \mathbf{0}$ is both necessary and sufficient for the convergence of Algorithm 1. If Algorithm 1 does converge, then the iterates $\{p_n^k\}_{k\geq 0}$ admit a limit $\lim_{k\to\infty} p_n^k \rightsquigarrow p_n^*$ where $p_n^*$ lies on the solution manifold of (2).*

Theorem 1 rigorously characterizes and clarifies the underlying relationships among several commonly employed stopping conditions, visually illustrating these connections in Figure 1. Theorem 1 serves as an essential foundation for our subsequent optimization studies. It is also worth noting that convergence of TMLE is a sufficient but *not* necessary condition for solving the Estimating Equation (2), which implies the potential occurrence of overshooting phenomena during algorithmic iterations (as discussed later in Section 4.4).

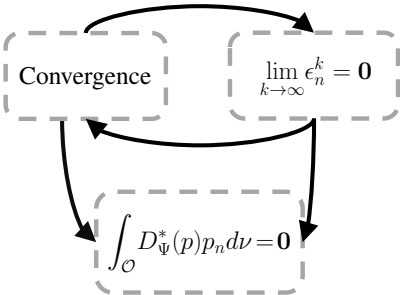

Figure 1: Illustration of Thm. 1.

## 4.2 Global convergence

The state-of-the-art convergence result regarding Algorithm 1 is currently represented by Lemma 1.

**Lemma 1.** *In Algorithm 1, the sequence of empirical risks $\left\{ \int_{\mathcal{O}} \mathbf{L}(p_n^k) p_n d\nu \right\}_k$ converges as $k \rightsquigarrow \infty$.*

*Proof.* See Section 3 in Van Der Laan and Rubin [5] for a detailed proof. □

However, it is a common consensus within the optimization community that convergence of the empirical loss alone does not necessarily imply convergence of the iterates themselves. This issue becomes even more challenging in a semi-parametric context.[8] We provide motivating examples of TMLE divergence without standard assumptions in Appendix I. Nonetheless, our analysis will build upon Lemma 1, and we first present several preparatory results essential to our analysis.

**Theorem 2** (Non-self-intersection). *The homotopy path $\{p(\epsilon) : \epsilon\}$ in the probability simplex never self-intersects for $d = 1$. Further assume the non-degeneracy condition*

$$\forall \beta \in \mathbb{R}^d \backslash \{\mathbf{0}\}, \quad \mathbb{P}\left\{ o \in \mathcal{O} : \beta^\top D_\Psi^*(p)(o) \neq 0 \right\} > 0. \tag{10}$$

*Then the $\epsilon \mapsto p(\epsilon)$ defines a one-to-one $\mathbb{C}^1$ embedding of $\mathbb{R}^d$ into the probability simplex, and its image is free of self-intersections for every $d \in \mathbb{Z}^+$.*

**Remark 1.** *The assumption for $d \geq 2$ is to require that the $d$ components of $D_\Psi^*(p)(o)$ are linearly independent in $L^2(\nu)$. Equivalently, $\Sigma = \int D_\Psi^*(p)(o) D_\Psi^*(p)(o)^\top p(o) d\nu(o)$ is a full-rank $d \times d$ matrix.[9]*

**Theorem 3** (Solution Submanifold). *Assume $n < \infty$ and the Fréchet derivative $\mathcal{D}_f D_\Psi^*$ is surjective for every $p \in \mathcal{M}$, then both the (i) IF Estimating Equation (2), and (ii) nonparametric loss landscape $\min_{\{p \in \mathcal{M}\}} \int_{\mathcal{O}} \mathbf{L}(p) p_n d\nu$ admit infinitely many solutions which form a smooth submanifold (or continuum) of a whole equivalence class in $\mathcal{M}$.*

We know that $\epsilon = \mathbf{0}$ implies $p(\epsilon) = p$, while Theorem 2 characterizes the converse direction, i.e., if $p(\epsilon) = p$ then $\epsilon = \mathbf{0}$. It also reveals a profound geometric structure underlying TMLE's iterative process. Specifically, the homotopy mapping induced by each fluctuation submodel defines a smooth embedding into the probability simplex. This embedding represents a continuous deformation of the statistical model that preserves its topological structure without self-intersections (cf. Figure 2), ensuring that each TMLE fluctuation traverses a well-defined homotopy path that cannot revisit the same density twice, preventing potential cycling behavior. Theorem 3 implies that the

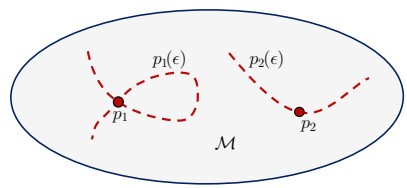

Figure 2: An illustration of self-intersection in $\mathcal{M}$. The path $p_1(\epsilon)$ (left) is self-intersect while the other $p_2(\epsilon)$ (right) is not.

solution submanifold is inherently infinite-dimensional, consisting of uncountably many solutions. However, it forms a well-defined, smooth "sheet" of equivalent solutions that simultaneously solve the estimating equation and the (approximate) empirical-risk minimization problem. Inspired by Theorem 3, we can geometrically interpret the iterative optimization procedure of TMLE on the

---

[8]In this context, "semi-parametric" refers to settings where the target parameter is defined parametrically, but the plug-in distribution is estimated using nonparametric methods.

[9]Since $\beta^\top \Sigma \beta = \int p(\beta^\top D_\Psi^*(p))^2 d\nu$, where by assumption the integrand is strictly positive in part of domain.

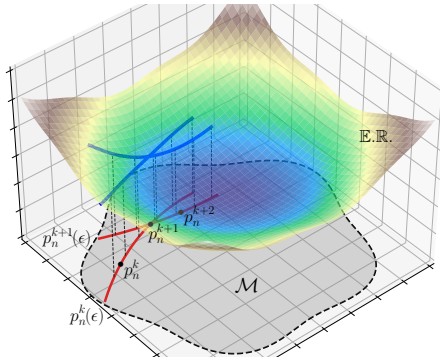
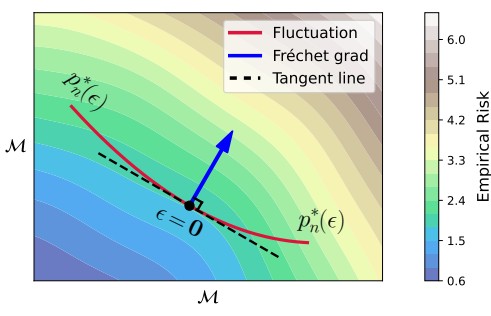

(a) Understanding of iterative process.  (b) Characterization of limiting distribution.

Figure 3: The conceptual diagram of `TMLE` procedure. In 3a, the gray region represents model $\mathcal{M}$ and $\mathbb{E.R.}$ denotes the empirical risk. At $k$-th iteration, we span a submodel (red curve) around $p_n^k$ in $\mathcal{M}$, then project the path onto the loss landscape (blue curve). We select the point corresponding with the minimal loss as $p_n^{k+1}$. 3b is the top-down perspective of 3a, which indicates the Fréchet gradient of loss evaluated at the $p_n^*$ is orthogonal (in the $L^2$ sense) to the tangent line of the submodel at $\epsilon = \mathbf{0}$.

empirical loss landscape (cf. Figure 3a), as well as characterize the geometric structure of the limiting distribution (cf. Figure 3b).

Grounded in the insights from Theorems 2 and 3, our asymptotic convergence guarantees are now provided in Theorem 4, demonstrating the infinite-step convergence behavior of `TMLE` even when the initial estimator is heavily misspecified in $\mathcal{M}$.

**Theorem 4** (Iterative Convergence). *Let $\{p_n^k\}_{k \geq 0}$ be the sequence of density estimates produced by the `TMLE` Algorithm 1. There exists a limiting density $p_n^* \in \mathcal{M}$ such that $\lim_{k \to \infty} p_n^k \rightsquigarrow p_n^*$.*

**Remark 2.** *Theorem 4 covers many well-known `TMLE` instances like Díaz and Rosenblum [25]. Note that even if Assumption 5 may not hold for some `TMLE` practices, we can still establish asymptotic regularity (or pseudo-convergence) of the iterates as $\lim_{k \to \infty} \|p_n^{k+1} - p_n^k\|_1 = 0$ (a.k.a., quasi-Cauchy sequence).*

We emphasize that the proof of Theorem 4 does not depend on the quality of the initial estimate $p_n^0$, meaning that Algorithm 1 guarantees asymptotic convergence to a solution of the *estimating equation* from *any* starting point (Theorem 1). However, a poor initial estimate can slow the decay of the empirical process term and the second-order remainder in Eq. (1), preventing the estimator from achieving parametric-rate efficiency asymptotically [6].

### 4.3 One-step property

**Theorem 5** (One-step Property). *The semi-parametric `TMLE` procedure performs exactly one update when one of the following conditions is met:*

(i) *initial density $p_n^0$ already on the solution manifold of (2) and $\int_{\mathcal{O}} \mathbf{L}$ is strictly convex in $\epsilon$;*

(ii) *for $\forall o \in \mathcal{O}$, the mapping $\epsilon \mapsto D_{\Psi}^*(p(\epsilon))(o)$ is a conservative (i.e., curl-free) vector field in $\epsilon$ and the loss satisfies line integral $\mathbf{L}(p(\epsilon))(o) \triangleq \mathbf{L}(p(\mathbf{0}))(o) + A \int_0^{\epsilon} D_{\Psi}^*(p(u))(o)du$;*

(iii) *in some of the practical problems on outcome regression (e.g., ATE, propensity-score intervention) with the existence of "clever covariate" and a proper fluctuation submodel;*

(iv) *hit the user-set convergence criteria (e.g., machine precision, number of iterations).*

In the (*i*), (*ii*) of Theorem 5, we firstly establish a set of sufficient conditions under which the `TMLE` can be terminated after one single iteration. Putting together with existing conditions (*iii*) and (*iv*), the whole theorem significantly extends various scattered conditions previously dispersed throughout existing literature, e.g., [42, 43, 25, 44]. It is important to clarify that this one-step property does

*not* imply convergence in the mathematical sense; instead, it indicates the algorithm has achieved a predefined stopping criterion (e.g., satisfying Estimating Equation (2)) and obtained desirable properties for the targeted estimator of interest.

### 4.4 Potential overshooting

As a by-product of our analysis into the optimization, we uncovered an interesting phenomenon wherein a `TMLE` update may overshoot a feasible solution to the *estimating equation* along the homotopy path induced by the fluctuation submodel (cf. Figure 4). We establish theoretical existence of this overshooting phenomenon through a concrete illustrative example presented in Example 3.

**Example 3** (Degenerate Hyperplane). *Consider Submodel (4) with log-likelihood loss. Solving*

$$\int_{\mathcal{O}} D_{\Psi}^*(p_n^k)(o) \left(1 + \epsilon^\top D_{\Psi}^*(p_n^k)(o)\right)^{-1} p_n d\nu(o) = \mathbf{0} \tag{11}$$

*with $\epsilon \neq \mathbf{0}$ gives the next movement of `TMLE`. If all of the $D_{\Psi}^*(p_n^k)(O_i)$ happen to lie in a single affine hyperplane $\{x : \epsilon^\top x = \chi\}$ where $\chi \in \mathbb{R}$ is a constant, the resulting distribution would exactly solve the Estimating Equation (2) while Algorithm 1 keeps looping.*

Motivated by Example 3, we present a more formal definition of this phenomenon.

**Definition 4** (Overshooting of TMLE). *At the $k$-th iteration of `TMLE`, we say that the Algorithm 1 overshoots a feasible solution if there exists $\epsilon^\dagger \neq \epsilon_n^k$ such that $p_n^k(\epsilon^\dagger) \in \mathcal{M}$ and $\int_{\mathcal{O}} D_{\Psi}^*(p_n^k(\epsilon^\dagger))(o) p_n d\nu(o) = \mathbf{0}$.*

> **Theorem 6** (Overshoot Control). *If we further assume that*
>
> (i) *the population Jacobian $\int_{\Omega} p_0 \nabla_\epsilon D_{\Psi}^*(p_n^k(\epsilon))\big|_{\epsilon=\mathbf{0}} d\nu$ has positive minimal eigenvalue $\lambda_o$,*
>
> (ii) $\left\| \int_{\mathcal{O}} p_n \left[ \nabla_\epsilon \mathbf{L}(p_n^k(\epsilon)) - \nabla_\epsilon \mathbf{L}(p_n^k(\mathbf{0})) \right] d\nu - \int_{\mathcal{O}} p_n \nabla_\epsilon^2 \mathbf{L}(p_n^k(\epsilon))\big|_{\epsilon=\mathbf{0}} d\nu \cdot \epsilon \right\| \lesssim \|\epsilon\|^2.$
>
> *Then, the probability that `TMLE` overshoots (o.s.) the nearest root of the* estimating equation *is*
>
> $$\mathbb{P}\left[o.s.\right] \lesssim 2d \exp\left(-\frac{n\lambda_o^2 C_o^2 \tilde{\mu}}{2B_o^2}\right) + 2d \exp\left(-\frac{n\tilde{\mu}}{2B_o^2}\right), \quad \tilde{\mu} = \left\| \int_{\Omega} p_0 D_{\Psi}^*(p_n^k) d\nu \right\|_\infty^2, \tag{12}$$
>
> *where $B_o$, $C_o$ are constants specified in Appendix H.*

While a systematic characterization of the exact conditions leading to overshooting remains an open challenge, we derive preliminary probabilistic guarantees under stronger assumptions in Theorem 6, particularly the uniform boundedness of second-order remainders, as specified in Condition *(ii)* above. Our findings indicate that the exponent in (12) scales linearly with sample size $n$, leading to *an exponential reduction in the overshooting probability bound as $n$ increases*. Additionally, *higher*-dimensional $\Psi$ *increases* the likelihood of overshooting and complicates the convergence. Intuitively, when $d \geq 2$, the solution set forms a complex manifold, where the risk function may ex-

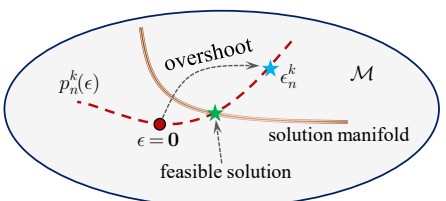

Figure 4: Illustration of an overshooting where the `TMLE` update passes through an feasible EIF-root ★ but continues to the next iterate ★.

hibit heterogeneous curvature, flat in some directions while sharply curved in others. Consequently, optimizers following steepest descent directions may escape the manifold before reaching a risk minimum, overshooting potential solutions. Furthermore, common subproblem solvers employing backtracking line searches or momentum acceleration are particularly prone to this phenomenon. The combination of adaptive step sizing and inertia can cause the algorithm to leap over viable roots before the termination criteria are triggered.

## 5 Proof Overview

This section sketches the high-level ideas behind our proofs.

***Proof Sketch of Theorem 1.*** This theorem naturally decomposes into 4 independent sub-results, which we denote by E1, E2, E3, and E4 (see Figure 5). For E1, we begin by showing that the sequence $\{p_n^k\}_{k \geq 0}$ enjoys a tail-sum additivity property in the underlying metric space. By carefully bounding the increments and summing over the fluctuation steps, we conclude that $\{p_n^k\}$ is a Cauchy sequence. In proving E2, we derive both compact upper and lower bounds on the fluctuation parameters $\{\|\epsilon_n^k\|\}$. Applying the squeeze theorem we deduce that the norm of $\epsilon_n^k$ must converge to the origin. Sub-result E3 follows directly from the classical analysis in Van Der Laan and Rubin [5], and it is worth noting that the proof remains straightforward under our framework. Lastly, sub-result E4 emerges as a trivial generalization of sub-result E3. □

***Proof Sketch of Theorem 2.*** Since the fluctuation submodel is defined via an injective link on its effective domain, in the scalar case ($d = 1$) we show that the Hellinger distance between $p(\epsilon_1)$ and $p(\epsilon_2)$ never vanishes. Hence $\epsilon \mapsto p(\epsilon)$ is strictly monotonic in the simplex. For general $d$, the non-degeneracy condition (10) guarantees that $\mathcal{D}_f p(\epsilon)[h] \neq 0$ whenever $h \neq 0$. One shows that $\|\epsilon_1 - \epsilon_2\| \lesssim \|p(\epsilon_1) - p(\epsilon_2)\| \lesssim \|\epsilon_1 - \epsilon_2\|$, thus $p$ is a bi-Lipschitz embedding of $\mathbb{R}^d$ into the simplex and cannot self–intersect by Hadamard's global inverse function theorem. □

***Proof Sketch of Theorem 3.*** We first show that the functional $h \mapsto \int_{\mathcal{O}} D_\Psi^*(p_0 + h)(o)p_n d\nu(o)$ is locally Lipschitz-type continuous in an $L^2(\nu)$-neighborhood of the reference solution $p_0$. Using a second-order Fréchet expansion we derive $\| \int_{\mathcal{O}} (D_\Psi^*(p_0 + h_1) - D_\Psi^*(p_0 + h_2))p_n d\nu \|_{\mathbb{R}^d} \lesssim \|h_1 - h_2\|_{L^2(\nu)}$. Given that the Fréchet derivative $\mathcal{D}_f D_\Psi^*(p_0)$ is assumed surjective, the rank–nullity theorem for Banach spaces ensures that the null space must be infinite-dimensional. The infinite-dimensional implicit function theorem then applies due to continuity and surjectivity conditions, which guarantees that the solution locus is a $\mathbb{C}^1$ manifold. The proof of *(ii)* technically follows a similar process. □

***Proof Sketch of Theorem 4.*** Along the real-analytic fluctuation path, we first show that $\|\epsilon_n^k\|^2 \lesssim \int_{\mathcal{O}} (\mathbf{L}(p_n^k)(o) - \mathbf{L}(p_n^{k+1})(o))p_n d\nu(o)$ using Lipschitz-in-path. Based on Lemma 1, telescoping this inequality shows that the squared step sizes form a summable series $\sum_{k=0}^{\infty} \|\epsilon_n^k\|^2$. Therefore, we get $\epsilon_n^k \rightsquigarrow \mathbf{0}$, combining these facts with the equivalence statement in Theorem 1, the vanishing of $\epsilon_n^k$ is both necessary and sufficient for convergence of the TMLE iterates. □

***Proof Sketch of Theorem 5.*** For case *(i)*, the result follows directly by substituting into the algorithmic framework and exploiting convexity arguments. In the analysis of *(ii)*, we leverage fundamental properties of line integrals along with optimality conditions to demonstrate that the updated density after one iteration lies within a component of the solution manifold. For case *(iv)*, we rigorously establish both the existence and appropriateness of stopping conditions introduced therein. We omit the proof for *(iii)*, as it is highly problem-specific and can be found in the corresponding paper. □

***Proof Sketch of Theorem 6.*** We first linearize the empirical score map at the origin, writing it as a fixed Jacobian term plus a Lipschitz remainder. The minimal-norm root therefore lies within a factor of the score norm, and a single loss-based update stays in the same radius. Overshoot can happen if the score at the origin is already large compared with that radius. Each coordinate of that score is a bounded average, by applying Hoeffding's inequality twice we obtain the desired bound. □

**Remark 3.** *We also provide several toy examples to validate partial results in Appendix B.1.*

## 6 Discussions

In this work, we have laid a rigorous optimization-theoretic foundation for TMLE. Our analysis assumes exact solution of each subproblem, whereas in numerical practice one always computes an approximate $\hat{\epsilon}_n^k$ satisfying $\|\hat{\epsilon}_n^k - \epsilon_n^k\|_2 \leq \sigma_\epsilon$ and setting $p_n^{k+1} \leftarrow p_n^k(\hat{\epsilon}_n^k)$. How such a gap affects convergence guarantees remains unclear. Meanwhile, the non-self-intersection property of submodel paths plays a subtle role and may carry implications for algorithmic stability. Further, our framework addresses only canonical first-order TMLE while extending to higher-order variants [28] is also an important directions for future research.

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
