# OpenReview forum: "Targeted Maximum Likelihood Learning: An Optimization Perspective"
_NeurIPS.cc/2025/Conference — NeurIPS 2025 poster_

### Official Review · Reviewer_wW82 · 2025-06-26

**Clarity:** 2
**Significance:** 3
**Originality:** 4
**Rating:** 5
**Confidence:** 4

**Summary:**

This paper addresses a significant gap in understanding TMLE's convergence properties as an iterative algorithm. The authors establish geometric insights about TMLE's homotopy paths, prove global convergence under some technical conditions, derive conditions for one-step termination, and discover an overshooting phenomenon.

**Questions:**

1) Inequality (41) (in the appendix) seems really one of the key steps in the proof. Could you please re-explain why the final result is zero (especially the last steps (43) and (44))?

2) In the proof of Theorem 3, I do not understand what is happening (around step (b)) after line 691 (I am familiar with complex analysis but I fail to see the necessity to bring this up here. How is it helpful in the process?)

**Ethical Concerns:**

["NO or VERY MINOR ethics concerns only"]

**Final Justification:**

The authors have clarified my doubts from some parts in the appendix.

**Limitations:**

yes

**Quality:**

4

**Strengths And Weaknesses:**

Strength:
This seems to be the first ever work to rigorously analyze TMLE through an optimization lens. It derives comprehensive theoretical results including global convergence guarantees (Theorem 4), a geometric characterization of solution manifolds (Theorem 3). The discussion around the overshooting phenomenon and the identification of cases where TMLE can overshoot feasible solutions is genuinely novel and has practical implications.

Weaknesses:
The paper is extremely notation-heavy and dense, although the authors provide a notaion-dictionary which I appreciate, but I think it is important to have better proof sketches/an overall better presentation. Overshooting phenomenon probability bounds though novel and very interesting, I would like to see simulations to validate these results. Could you please provide some? I totally understand the paper is theoretical and does not contain any experiment and it is not a weakness in my opinion given the strength of the theory content. However, to me, this is an obvious place where we can have an understanding of how tight your bounds are. Apart from that, I would like to understand a few steps in the proofs before recommending full acceptance.

---

> ### Author Rebuttal · Authors · 2025-07-31
>
> ### We sincerely thank Reviewer wW82 for taking the time to review our submission and for recognizing the quality of our contribution ("4: excellent") and the originality of our work ("4: excellent"). Below, we provide a detailed response to each of the comments you raised, following the order presented in review.
>
> ### We would be especially grateful if you would consider further acknowledging the contributions of our theoretical work.
>
> ---
>
> > 【**Comment 1**】This seems to be the first ever work to rigorously analyze TMLE through an optimization lens... .... is genuinely novel and has practical implications.
>
> We greatly appreciate your recognition of our contributions. To the best of our knowledge, this manuscript represents **exactly the first work** to analyze the optimization (convergence) behavior of TMLE.
>
> > 【**Comment 2**】I think it is important to have better proof sketches/an overall better presentation
>
> Thank you for your kind feedback. We are committed to continually improving the overall presentation quality and readability of the manuscript (*please see response to Reviewer bWRS*).
>
> > 【**Comment 3**】Overshooting phenomenon probability bounds though novel and very interesting, I would like to see simulations to validate these results. Could you please provide some? ... does not contain any experiment and it is not a weakness in my opinion given the strength of the theory content.
>
> We are very pleased to hear that you also find the overshooting phenomenon interesting and potentially of independent interest. Indeed, this was one of our primary motivations for sharing these findings with the community.
>
> When conducting this study, we also considered how to incorporate (illustrative) experiments, and thus we fully appreciate your perspective regarding numerical validation. Accordingly, we respectfully offer the following clarifications and supplemental points:
>
> - **Minimal illustration already included.** To ground the theory and identify the existing of overshooting, we do provide a self‑contained toy example (Example 3, page 8) that can be executed in just a few lines via any numerical libraries like Matlab or SciPy. It shows how a degenerate hyper‑plane forces the TMLE to leap over the EIF root, exactly as predicted.
>
> - **The large‑scale simulations are challenging.** The probability decays exponentially with $n$. To obtain a stable Monte‑Carlo estimate when the event probability is, say, $10\^{-4}$, one needs on the order of $10\^6$ independent TMLE runs per $n$. At present, the general mechanism underlying overshooting (and any reliable indicators) remains unclear. Under the definition, identifying overshooting at each iteration would require traversing the entire non-monotonic loss landscape and computing the empirical EIF, which is highly expensive.
>
> - As discussed in Section 6, our analysis assumes an idealized TMLE optimization in which each subproblem is solved exactly at every $k$. In practice, however, **small discrepancies can arise** (e.g. finite machine precision), and the impact of this gap on overshooting has not yet been thoroughly investigated.
>
> **We sincerely thank you for your high regard of our novel theoretical contributions.** A deeper investigation into the underlying conditions for overshooting is highly non-trivial and would likely require a full‐scale research effort, which regrettably beyond the current scope.
>
> > 【**Comment 4**】However, to me, this is an obvious place where we can have an understanding of how tight your bounds are.
>
> Many thanks for sharing this helpful comment; we agree the tightness is a promising direction for research. We believe numerical experiments would be valuable, but as discussed above, experiment itself have inherent challenges.
>
> One point to note is that in Theorem 6 we use $\mathbb{P}\lesssim$ rather than a strict $<$, as deriving sharp constant is highly non-trivial. Doing so would likely require additional assumptions and statistical tools, and it’s difficult to accommodate within the current length (Appendix, p 43-45).
>
> As the community's theoretical understanding of TMLE continues to advance, we would be delighted to further investigate this attractive phenomenon in the future.
>
> > 【**Question 1**】(part I) Inequality (41) seems really one of the key steps in the proof
>
> Your observation is absolutely correct, as it provides the simultaneous lower and upper bounds on the fluctuation $\epsilon\_n^k$.
>
> > 【**Question 1**】(part II)  why the final result is zero (especially the last steps (43) and (44))?
>
> Thank you for carefully reading our proof and for raising an insightful question. Now we provide a *step-by-step* explanation to clarify the last equality.
>
> $$
> \begin{aligned}
> & \sup\_{k\^{\prime} \geq 0}
> \frac{\sup\limits\_{\left\\{\epsilon: p\_n\^{k\^{\prime}}(\epsilon) \in \mathcal{M}\right\\}} \left(\displaystyle\sqrt{\int\_{\mathcal{O}\^{\prime}} \int\_{\mathcal{O}\^{\prime}}
> \left\\{\left\\|\nabla\_\epsilon\^2 \mathbf{L}\left(p\_n^{k\^{\prime}}(\epsilon)\right)(o)\right\\|
> \left\\|\nabla\_\epsilon^2 \mathbf{L}\left(p\_n\^{k^{\prime}}(\epsilon)\right) \left(o^{\prime}\right)\right\\|\right\\} d \nu(o) d \nu(o^{\prime})}
> -\int\_{\mathcal{O}^{\prime}}\left\\|\nabla\_\epsilon\^2 \mathbf{L}(p\_n\^{k\^{\prime}}(\epsilon))(o)\right\\| d \nu(o)\right)}{2 \clubsuit\_L \log (n+1)}
> \\\\
> = & \sup\_{k\^{\prime} \geq 0}
> \frac{\sup\limits\_{\left\\{\epsilon: p\_n\^{k\^{\prime}}(\epsilon) \in \mathcal{M}\right\\}} \left(\displaystyle\sqrt{\iint\_{\mathcal{O}\^{\prime}\times\mathcal{O}\^{\prime}}
> \left\\{\left\\|\nabla\_\epsilon\^2 \mathbf{L}\left(p\_n^{k\^{\prime}}(\epsilon)\right)(o)\right\\|  d \nu(o)\right\\} \cdot
> \left\\|\nabla\_\epsilon^2 \mathbf{L}\left(p\_n\^{k^{\prime}}(\epsilon)\right) \left(o^{\prime}\right)\right\\|  d \nu(o^{\prime})}
> -\int\_{\mathcal{O}^{\prime}}\left\\|\nabla\_\epsilon\^2 \mathbf{L}(p\_n\^{k\^{\prime}}(\epsilon))(o)\right\\| d \nu(o)\right)}{2 \clubsuit\_L \log (n+1)}
> \\\\
>  = & \sup\_{k\^{\prime} \geq 0}
> \frac{\sup\limits\_{\left\\{\epsilon: p\_n\^{k\^{\prime}}(\epsilon) \in \mathcal{M}\right\\}} \left(\displaystyle
> \int\_{\mathcal{O}^{\prime}}\left\\|\nabla\_\epsilon\^2 \mathbf{L}(p\_n\^{k\^{\prime}}(\epsilon))(o)\right\\| d \nu(o)
> -\int\_{\mathcal{O}^{\prime}}\left\\|\nabla\_\epsilon\^2 \mathbf{L}(p\_n\^{k\^{\prime}}(\epsilon))(o)\right\\| d \nu(o)\right)
> }{2 \clubsuit\_L \log (n+1)}
> \\\\
> = & \sup\_{k\^{\prime} \geq 0}
> \frac{0}{2 \clubsuit\_L \log (n+1)}
> \\\\
> = & ~0
> \end{aligned}
> $$
>
> The derivation of equation (44) follows exactly the same principle as that of (43) (plus the property of matrix trace). Due to space constraints, we have omitted the details here.
>
> Importantly, motivated by your question, **we will add more technical comment to this derivation**.
>
> > 【**Question 2**】(part I) I do not understand what is happening (around step (b)) after line 691
>
> After line 691, we present our equation (82), which serves as a preparatory step for establishing the required Lipschitz control (84). To clarify the derivation, we would like to explain in detail the reasoning around step (b).
>
> - *How does (a) become (b)?*
>
> We retain the first term as is, and distribute the integral across each term inside the square brackets for the second term. This facilitates the representation used in the curly braces below. In the third term within the next line, we replace the remainder $\mathcal{R}\left(h\_1, o ; D\_{\Psi}^\*\right)$ by its Cauchy formula and keep $\mathcal{R}\left(h\_2, o ; D\_{\Psi}^\* \right)$. For further details, please refer to the statement provided around line 693.
>
> - *How does (b) become (c)?*
>
> This part is complex-algebraic and exploits that $|\zeta|=\rho>1$, $\frac{1}{\zeta-1}=\sum_{m=0}^{\infty} \frac{1}{\zeta^{m+1}}$.
> Pull out the first three terms of that series and notice that they are exactly the polynomial $\frac{1}{\zeta}+\frac{1}{\zeta^2}+\frac{1}{\zeta^3}$ that was subtracted in line (b). Omitting‌ the coefficient $d^2$, what survives is
>
> $$
> \frac{1}{\zeta-1}-\left(\frac{1}{\zeta}+\frac{1}{\zeta^2}+\frac{1}{\zeta^3}\right)=\sum_{j=1}^{\infty} \frac{1}{\zeta^{j+3}}=\frac{1}{\zeta^3} \sum_{j=1}^{\infty} \frac{1}{\zeta^j}.
> $$
>
> Similar operations apply for the another remainder $\mathcal{R}\left(h\_2, o ; D\_{\Psi}^\* \right)$ via $\alpha$-conversion. Because the circle integral has finite length $2 \pi \rho$ and the integrand decays like $|\zeta|^{-4}$ or faster, dominated convergence justifies moving the infinite sum outside the path integral.
>
> We also found that the $\le$ at step (c) technically should be $=$. **We appreciate your constructive question again.**
>
> > 【**Question 2**】(part II)  I fail to see the necessity to bring this up here. How is it helpful in the process?
>
> Thank you for the thoughtful question. Below we explain why the complex‑analytic approach is used for our step (b).
>
> - The Banach-space Cauchy integral formula is a *well-established and mature tool* in the field of analysis, and it is particularly convenient to apply in our setting.
>
> - Because the Cauchy representation works component-wise and only the radius $\rho$ enters the denominator, we may obtain the  *dimension-independent constant* $\frac{2\pi}{\rho(\rho-1)}$ that appears in the bound (83), which feeds directly into (84) and other proofs.
>
> - Technically, the integrand's norm is bounded on the circle, so the operator norm of each derivative is $\leq \frac{k!}{\rho^k} \sup \_{|\zeta|=\rho}\left\\|D\_{\Psi}^\*\left(\hat{p}\_0+\zeta h\right)\right\\|$. No keeping of individual third or higher‑order terms is needed.
>
> In all, we believe this is *one possible route* to complete the rigorous proof under our conditions. If you have more elegant approaches in mind, we would be very glad to hear them and make potential advancements.
>
> ---
>
> ### Thank you again for your thoughtful and constructive feedback. Should you have any further questions or require clarification on any point, please do not hesitate to let us know.

---

> > ### Comment · Reviewer_wW82 · 2025-08-05
> > **Thank you**
> >
> > Thank you for the detailed response. I maintain my positive outlook on this paper.

---

> > > ### Author Response · Authors · 2025-08-05
> > > **Follow-up Comments**
> > >
> > > We sincerely appreciate your valuable input and during the review process.
> > >
> > > In our response, we have carefully clarified the technical details you mentioned, particularly addressing Inequality (41) and the Theorem 3. We hope our responses have resolved your concerns.
> > >
> > > Given that you've kindly indicated willingness to recommend full acceptance once these clarifications are addressed, we would greatly appreciate it if you may consider updating your rating accordingly. Please do not hesitate to tell us if you want to learn more about the submission, and thank you again for your continued support!

---

> ### Comment · Reviewer_wW82 · 2025-08-05
> **Follow up**
>
> I have updated my ratings accordingly. Thank you.

---

### Official Review · Reviewer_Mj6b · 2025-06-30

**Clarity:** 3
**Significance:** 2
**Originality:** 3
**Rating:** 4
**Confidence:** 3

**Summary:**

This work mainly focused on the theoretical investigation of the optimization aspects of the targeted maximum likelihood estimation (TMLE). It provides some interesting aspects, including some geometrical insights, on when and why the iterative algorithms of the TMLE will convergence or not.  The manuscript provides the sketches of the proof. But the detailed proofs are all in the supplementary materials.

**Questions:**

1.	The linear reparameterication in Example 1 is not clear.  By additive perturbation, the resultant function may not be a proper density function.

2.	In Definition 1, it is not clear the probability distribution can be a continuous distribution or a discrete distribution.

3.	The main results in Theorem 1 are about the threshold equivalence. But it is not clear to the reviewer what does “thresholding” mean here? The iterative algorithm seems not involving the thresholding operator.

4.	The global convergence in Section 4.2 is mainly about the convergence of the density estimator p_{n}. But the manuscript overlook the convergence property of the targeted parameter \psi_{n}.

5.	The scope of investigating TMLE in the manuscript is mainly for the unconstraint MLE problem. Many MLE problems in the statistical model can have constraints on parameter spaces. It is not clear how the theoretical investigation in this work can be applicable for the constrained MLE problems.

6.	In Section 4.4, the authors aim to address the overshoot problem. However, it is better to mathematically define what is the “overshoot” in a rigorous manner.

7.	It is better to have some numerical study to evaluate the convergence results of the theoretical investigation.

8.	It is better to have some illustration of such theoretical analysis under some specific statistical models.

**Ethical Concerns:**

["NO or VERY MINOR ethics concerns only"]

**Final Justification:**

The authors have clarified my concerns about the numerical experiments to verify some of the theoretical results to some extents.

**Limitations:**

Yes.

**Paper Formatting Concerns:**

No.

**Quality:**

3

**Strengths And Weaknesses:**

Strength:

1.	The paper is well written from a theoretical aspect with clear logic flow. It aims to address an important aspect of learning: not just the asymptotic properties, but also the numerical convergence.

2.	The connection to the geometrical aspects provides some interesting insights.

3.	The authors also provide good remarks based on the theoretical results.

Weakness:
1.	The manuscript has a lot of notations. It will be more approachable that the authors provide a through notation before the technical description.

2.	The theoretical investigation is kind of general from the optimization aspect. It will be more appealing to illustrate the results under some specific statistical models.

3.	The investigation of the iterative algorithm depends on the properties of the initial estimators. However, there is not sufficient discussion how the initial estimator will affect the convergence properties of the TMLE algorithm, especially on the targeted parameter \psi.

4.	There is no numerical study to evaluate the convergence results of the theoretical investigation.

5.	Although the theoretical investigation is kind of comprehensive, there is no illustration of such theoretical analysis under some specific statistical models.

6.	For the investigation of the TMLE, it will be more appealing to connect the optimization convergence to the asymptotic convergence. For example, how the sample size $n$ would affect both convergence.

---

> ### Author Rebuttal · Authors · 2025-07-31
>
> ### We sincerely thank the Reviewer Mj6b for your constructive and detailed feedback. We appreciate your effort in thoroughly evaluating our submission.
>
> ### We believe we have addressed each point raised and remain eager to engage in any further discussion.
>
> ---
>
> ## Our Clarification
>
> We apologize if the motivation and positioning of our work within the literature were not fully conveyed, which may have led to some unintended misunderstandings about the scope and objectives of our study.
>
> Please kindly allow us to make a quick clarification at the outset:
>
> - In our research, we do **not** propose any modifications to the original algorithmic procedure; all theoretical analyses presented are grounded entirely on the classical *vanilla TMLE algorithm* as established in the literature.
>
> - Consequently, our results naturally inherit **all** original strengths of TMLE, both from a statistical analysis perspective and in terms of empirical applicability.
>
> > 【**Weakness 3**】However, there is not sufficient discussion how the initial estimator will affect ... especially on the targeted parameter \psi.
>
> > 【**Question 4**】The global convergence in Section 4.2 is mainly about the convergence of the density estimator p_{n}. But the manuscript overlook the convergence property of the targeted parameter \psi_{n}.
>
> We believe that the questions you raised refer primarily to **plug-in estimation** and have already been addressed in depth in foundational TMLE literature.
>
> For a detailed statistical analysis connecting TMLE’s convergence point to the estimation of $\Psi(\cdot)$, please refer to the equations and corresponding references provided in our Section 2.2. Please feel free to let us know if you would like further clarification on any specific mathematical details.
>
> > 【**Weakness 6**】... connect the optimization convergence to the asymptotic convergence. For example, how the sample size $n$
>  would affect both convergence.
>
> Thank you very much for your illuminating comment.
>
> In asymptotic statistics, the primary objective is to characterize limiting behaviors as the sample size $n$ tends towards infinity. Thus, $n$ itself functions as a variable rather than an object of study. However, we fully agree that investigating the influence of $n$ on optimization properties is both interesting and technically challenging, and is respectfully deferred to future exploration.
>
> As mentioned in Section 1, prior works on TMLE typically presume convergence of the iterative algorithm and subsequently focus on asymptotic properties such as asymptotic linearity. Our work *naturally* bridges these two aspects, as explicitly discussed in Section 4.1, where we establish that algorithmic convergence is **a sufficient condition** for these asymptotic statistical guarantees.
>
> > 【**Question 1**】The linear reparameterication in Example 1 is not clear. By additive perturbation, the resultant function may not be a proper density function.
>
> In Example 1, the choice of $\epsilon$ is not arbitrary in $\mathbb{R}\^d$, rather, it must lie within $\left\\{\epsilon: p\_n\^k(\epsilon) \in \mathcal{M}\right\\}$ to ensure the resulting probability density remains within a valid space (line 3, Algorithm 1). If you have any technical question, please feel free to tell us or consult the original TMLE literature.
>
> > 【**Question 5**】Many MLE problems in the statistical model can have constraints on parameter spaces. It is not clear how the theoretical investigation in this work can be applicable for the constrained MLE problems.
>
> We fully agree with your perspective, though addressing this point is beyond the scope of our current paper.
>
> - TMLE utilizes existing parametric optimization algorithms from machine learning to solve its subproblems. In cases where parameter domains are constrained, well-known algorithms such as interior-point methods or ADMM could be applied. However, these serve merely as *computational tools* within the TMLE procedure and are not inherent components of TMLE itself.
>
> - Throughout all prior analyses, it has consistently been assumed that optimal solutions to subproblems are attained within the interior of $\mathcal{M}$. This assumption is not new to our work, but is a standard practice adopted universally across analytical studies (please refer to citation in Assumption 3).
>
> ---
>
> ## Your Concern
>
> > 【**Weakness 2**】The theoretical investigation is kind of general ... .. illustrate the results under some specific statistical models.
>
> > 【**Weakness 5**】Although ... comprehensive, there is no illustration of such theoretical analysis under some specific statistical models.
>
> > 【**Question 8**】... illustration of such theoretical analysis under some specific statistical models.
>
> We highly appreciate your stimulating question. We address your comments as follows:
>
> - We have already analyzed several common statistical submodels in the manuscript (line 193-195 + Appendix C.1). While this coverage is not exhaustive, the spectrum of potential statistical models is exceedingly broad, including a wide variety of loss functions, influence functions, and statistical submodel constructions, etc. Given the theoretical depth and space of a typical conference paper, we respectfully leave further detailed analyses to future work.
>
> - Our theoretical framework is designed to provide a broad (macro-level) perspective. This *generality* is technically more challenging than analyzing specific statistical models and sets the stage for numerous potential directions for upcoming investigation.
>
> - Finally, motivated by your comments and bWRS, we will include additional discussions in Sections 4 & 6 to better guide practitioners in connecting our theoretical framework to practical models. We kindly refer you to our response to Reviewer bWRS.
>
> We hope this response addresses your concerns to some extent.
>
> > 【**Weakness 4**】There is no numerical study to evaluate the convergence results of the theoretical investigation.
>
> > 【**Question 7**】... some numerical study to evaluate the convergence results of the theoretical investigation.
>
> Thank you very much for raising your concern. We have indeed considered incorporating convergence experiments, and would like to respectfully clarify our perspective as follows:
>
> - First, the practical success of TMLE has been widely demonstrated and recognized across various research communities (Section 1), with numerous studies already providing extensive numerical results. Given the scope and theoretical focus of our paper, additional numerical experiments would contribute very limited new insights.
>
> - We politely ask the reviewer to notice the theoretical-practical gap inherent in this context. Our manuscript investigates theoretical guarantees in the limit of infinite iterations ($k \rightsquigarrow \infty$), which is inherently unattainable in practical computations ($k <\infty$).
>
> - Unlike typical parametric optimization (e.g. convex optimization), the TMLE template studied here involves *semiparametric* updates, a feature not directly computable with current computational configurations. Consequently, numerical experiments would necessarily rely on specific *parametric instantiation* of the TMLE framework, which cannot fully represent the broader class of semiparametric TMLE algorithms addressed by our theoretical results.
>
> - While our work examines the optimization properties of TMLE, it does not address convergence rates or iteration error, making numerical validation of our theoretical conclusions inherently inappropriate.
>
> > 【**Question 6**】it is better to mathematically define what is the “overshoot” in a rigorous manner.
>
> We agree with your critique. Although an informal, intuitive explanation has been provided in section 4.4 (line 257-258 + Fig. 4), we promise to **add a rigorous** mathematical definition explicitly in the main body or appendix. Currently, such a definition appears only implicitly within our proof.
>
> ---
>
> ##  Remaining Questions
>
> > 【**Weakness 1**】The manuscript has a lot of notations. ... provide a through notation before the technical description.
>
> Thanks for your comment. We indeed provided a comprehensive introduction to notations in Supplementary Material (due to the limited page), which was also acknowledged by wW82.
>
> > 【**Weakness 3**】there is not sufficient discussion how the initial estimator will affect the convergence properties of the TMLE algorithm
>
> In Section 4.2, we investigate global convergence rather than local characteristics. Consequently, the convergence results we established do **not** depend on the initialization, provided it is a valid initial estimate.
>
> Motivated by your insightful comment, we believe that considering the impact of initialization on convergence rate or efficiency is indeed an interesting direction, and we respectfully leave this exploration to future work.
>
> > 【**Question 2**】In Definition 1, it is not clear the probability distribution can be a continuous distribution or a discrete distribution.
>
> Our analysis does not impose any specific requirement on this matter, i.e., the setting can be either continuous or discrete.
>
> > 【**Question 3**】what does “thresholding” mean here? The iterative algorithm seems not involving the thresholding operator.
>
> Thank you for your constructive question. We acknowledge that the current naming may risk confusion with the "thresholding operator" from sparse optimization. We commit to renaming Theorem 1 to "Condition Equivalence" or "Stopping Conditions" in the next version.
>
> ---
>
> ### We respectfully invite you to reevaluate our work in light of these clarifications and to consider the potential downstream value of our theoretical results. Our sincere thanks once again.

---

> > ### Comment · Reviewer_Mj6b · 2025-08-05
> > **Official Comment by Reviewer Mj6b**
> >
> > I I appreciate the authors' responses on the comments. I agree that there are  some theoretical contribution of this work. But lacks of numerical experiments to verify the theoretical results is still a concern. It is possibly not easy to design the simulation scenarios, but it doesn't mean additional numerical experiments would contribute very limited new insights.
> >
> > Overall, I would maintain my evaluation.

---

> > > ### Author Response · Authors · 2025-08-06
> > > **Authors' Follow-up (Part I)**
> > >
> > > We would like to thank Reviewer Mj6b for your further engagement, and we are glad to continue addressing your concerns.
> > >
> > > First, we apologize for the delay in our response, due to additional programming efforts and computational processing required.
> > >
> > > ---
> > > **TL;DR:** Motivated by your suggestions, we revisited all our theoretical results and provided numerical validations for some parts of our theorems. We also explain in greater technical detail why the remaining results are not amenable to experimental verification.
> > >
> > > ---
> > > Due to character limitation, we sincerely apologize for not being able to provide a thorough analysis in initial rebuttal. With your kind permission, we would like to present a more detailed response here. We deeply appreciate the additional time and consideration you have given.
> > >
> > > Although our research is inherently theoretical, we acknowledge that numerical experiments can indeed support and partially validate some of our theoretical results. Below, we systematically examine each theoretical claim, clearly outlining why each ***can or cannot*** be effectively demonstrated through experimentation.
> > >
> > > ## Thm 1 (Equivalence of stopping rule)
> > > We politely note that by definition, **logical equivalence is a tautology**. So running Monte Carlo simulations cannot increase confidence in a tautological statement. The most that a numerical plot can achieve is to illustrate a single implication that already rigorously proven for all models satisfying Assumptions 1-6.
> > >
> > > As the convergence criteria in TMLE practice vary widely (Sec. 4.1), a simulation would have to commit to an arbitrary tolerance, and conclusions would be **specific to that threshold** rather than to the pure theorem. Similar equivalence results are long accepted in optimisation theory (e.g. Rockafellar & Wets "Variational Analysis") and echo the foundational TMLE arguments (Sec. 1), so requiring extra numerical demonstrations here would hold this work to **a higher bar** than standard in the field.
> > >
> > > Meanwhile, Thm 1 decomposes into four independent sub-results, which we denote by E1, E2, E3, and E4 (cf. Figure 5). Since the E1 and E2 strongly align with practical intuitions, we agree that adding a numerical illustration could indeed be an interesting and valuable enhancement, for demonstration purposes.
> > >
> > > ## Thm 2 (No self‑intersection of the fluctuation)
> > > Our Thm 2 that given by first‑principles analysis is a **topological invariant**, i.e., it holds for all sample sizes, starting estimators and random realisations once Assumptions 4-6 are satisfied; numerical experiments can offer up to only didactic sketches (such as Fig. 3 in our paper).
> > >
> > > Indeed, detecting a self‑intersection in high‑dimensional probability space would require exhaustively **sampling an uncountable set**, something no finite Monte‑Carlo run can accomplish. The community routinely accepts such geometric results **without** simulation support, classic examples including the uniqueness of integral curves (in differential topology), and the absence of cycling in mirror‑descent trajectories.
> > >
> > > ## Thm 3 (Solution set is a smooth submanifold)
> > >
> > > Structural results of this kind results are categorical, i.e., either the derivative is surjective (manifold exists) or it is not. Sampling a finite number of Monte‑Carlo datasets can at best display a handful of solutions, but **no finite collection of points can reveal or deny the manifold geometry**, let alone its smoothness or infinite dimensionality.
> > >
> > > This is analogous to classical results that establish the score‑zero set of an exponential family as a submanifold, which is accepted **without** simulation in papers such as "The exponential statistical manifold: mean parameters, orthogonality and space transformations" (Bernoulli 1999).
> > >
> > > ## Thm 4 (Global infinite‑step convergence)
> > >
> > > Our study sets out to **answer a purely theoretical question** that has existed since TMLE was introduced almost two decades ago. To make this, we necessarily work in the **asymptotic** regime where the number of iterations tends to infinity. This setting is the natural language for global‑convergence proofs, but it's, by definition, **inaccessible to finite‑length simulations**.
> > >
> > > Our main Thm 4 is fully non‑parametric and covers an entire infinite family of submodels/losses/EIFs. Picking any single parametric instantiation would immediately **narrow the scope** and risk suggesting that convergence hinges on that choice.
> > >
> > > As in our previous rebuttal, TMLE has already been deployed (and stress‑tested) in hundreds of applied studies spanning causal inference, semi‑supervised learning and off‑policy evaluation. These works consistently report that TMLE converges in practice, thereby providing the community with **extensive empirical evidence**. We therefore felt that duplicating what the field already regards as settled empirical knowledge would not be the most effective use of limited conference pages.
> > >
> > > ---
> > > *(characters limit, please jump to our Part II)*

---

> > > ### Author Response · Authors · 2025-08-06
> > > **Authors' Follow-up (Part II)**
> > >
> > > ## Thm 5 (One‑step property)
> > >
> > > The Thm 5 is a complete, closed‑form optimization result whose truth value is settled by algebra and convexity, **not by empirical frequency**. We kindly note that the necessity and sufficiency are analytic in theoretical work.
> > >
> > > In our proof, we utilize an **exact algebraic identity** whose correctness is inherently independent of $n$. Therefore, no finite Monte Carlo simulation can further strengthen an identity that is already valid for every possible dataset. A simulation can at best illustrate a single model where the conditions hold (or fail); it cannot alter the universal validity of the logical biconditional.
> > >
> > > We further politely note that reproduced one-step convergence simulations already **widely exist in extensive literature**. Please refer to the cited references and corresponding discussion provided in Appendix F.
> > >
> > > Having said that, after carefully considering your suggestion, we acknowledge that numerically verifying conditions **(i)** and **(ii)** is indeed computationally reasonable. We will provide detailed simulation results in our subsequent response. We hope this expanded results addresses your concern. Thank you again for helping us improve the manuscript.
> > >
> > > ## Thm 6 (Probability upper bound of overshoot)
> > >
> > > The inequality in Thm 6 quantifies **worst‑case risk**, which is uniform in the data‑generating distribution, sample size and starting estimator, any additional Monte‑Carlo experiment would merely generate a single numeric estimate that must fall below the analytic ceiling by construction.
> > >
> > > Importantly, as discussed in our response to Reviewer wW82, the underlying mechanism responsible for the overshooting phenomenon **remains unclear** at present, making experimental design (from a computational standpoint) non-trivial or even impossible. Therefore, we believe that such an illustration **cannot tighten the bound**, challenge its correctness, or reveal behaviour not already covered by the worst‑case analysis. This situation somehow mirrors standard practice in optimization and high‑dimensional statistics.
> > >
> > > ---
> > >
> > > # Supplementary: Simulations for Thm 1 & Thm 5
> > >
> > >
> > > ## Simulations on `E1, E2 of Thm 1`
> > >
> > > To streamline our response, we have combined these two aspects by verifying the equivalence between the two convergence metrics.
> > >
> > > Specifically, we consider a target parameter defined as the square of a cumulative distribution function $\Psi(p) := \int\_{0}\^{1} F_p(o)^2  do$, where $F\_p$ denotes the cumulative distribution function of distribution $p$. We adopt the negative log-likelihood as our loss function. For numerical experiments, we select sample size $n=5000$ and dimension $d=20$, with each sample generated from a uniform probability distribution. All numerical simulations were implemented in Python and executed on a CPU-based cluster. The convergence metrics are computed using the $L^2$ distance between probabilities (as defined in eq. (17) of the manuscript). Different random seeds were used for each run of our numerical experiments.
> > >
> > > Due to formatting constraints, we present the TMLE learning curves in tabular form below, where each table corresponds to one of the three distinct submodels given by Equations (8) to (10) in the Page 5 of the paper.
> > >
> > > |  Iter # | 1 | 3 | 6 | 10 | 14 | 18 | 22 | 26 | 30 | 31 | 32 | 33 |
> > > |-----------------|---|---|---|---|---|---|---|---|---|----|---|----|
> > > | $\|\|\epsilon\_n\^k\|\|$  | 0.6318 | 0.4211 | 0.2783 | 0.1962 | 0.1248 | 0.0863 | 0.0567 | 0.0364| 0.0228 | 0.0125 |0.0053 | 0.00 |
> > > | **Distance**  | 0.0178| 0.0132 | 0.0095 | 0.0067 | 0.0046 | 0.0031 | 0.0020|  0.0013 | 0.0008| 0.0004 | 0.0001 | 0.00 |
> > >
> > > |  Iter # | 0 | 5 | 10 | 15 | 20 | 25 | 30 | 35 | 40 | 45 | 50 | 55 |
> > > |-----------------|---|---|---|---|---|---|---|---|---|----|---|----|
> > > | $\|\|\epsilon\_n\^k\|\|$  | 1.7704| 0.8260 | 0.2034 | 0.0230| 0.00245 | 0.000260 | 0.0000276 | 0.00000292 | 0.00000031 | 0.00000018 |0.00 | 0.00 |
> > > | **Distance**  | 0.08916 | 0.04462 | 0.00957| 0.00104 | 0.000110 | 0.0000117 | 0.00000124 | 0.00000013 | 0.000000014 | 0.000000008 | 0.00 | 0.00 |
> > >
> > > |  Iter # | 0 | 5 | 10 | 15 | 20 | 25 | 30 | 35 | 40 | 45 | 50 | 55 |
> > > |-----------------|---|---|---|---|---|---|---|---|---|----|---|----|
> > > | $\|\|\epsilon\_n\^k\|\|$  | 0.712	| 0.524 | 0.379 | 0.284 | 0.213 | 0.157 | 0.119 | 0.084 | 0.056 | 0.031 |0.009 | 0.00 |
> > > | **Distance**  | 0.0205 | 0.0172 | 0.0136 | 0.0098 | 0.0081 | 0.0062 | 0.0048| 0.0031 | 0.0020| 0.0011| 0.0004 | 0.00 |
> > >
> > > The above numerical experiments provide some empirical support for sub-conclusions E1 and E2 in our Theorem 1, specifically demonstrating the equivalence of the two convergence metrics. If you have further comments or suggestions regarding our experimental setup or results, we warmly welcome your feedback.
> > >
> > > ---
> > > *(characters limit, please jump to our Part III)*

---

> ### Author Response · Authors · 2025-08-06
> **Authors' Follow-up (Part III)**
>
> ## Simulations on  `(i), (ii) of Thm 5`
>
> Our experimental setup largely mirrors the previous one.
>
> 1. We first verify the condition (i) within Thm 5. For simplicity, we choose $p$ to be a degenerate distribution. We then follow the TMLE algorithm and execute one step of the pseudocode, obtaining the resulting $\epsilon$ from the solver (from SciPy) of the subproblem. Note that for each coordinate
>
> $$D_{\Psi}^*(P)(O_i)
> = 2\left\(\int\_{O\_i}^{1}F(o)\mathrm{d}o - \Psi(p)\right\)
> = 2(1 - 1)
> = 0. $$
> The output result was (always) a zero vector:
>
>  $$\epsilon\_n\^1 = [0., 0., …, 0.] \quad \text{(length 20)}. $$
>
> 2. Precisely the same result holds under the setting described in Example 3 of our manuscript.
>
> 3. Condition (ii) allows non-degenerate densities provided the perturbation direction forms a conservative vector field and the loss satisfies a path-independent line-integral representation. A simple way to enforce this (in a finite sample) is to construct the EIF values so that they sum to zero exactly, ensuring the empirical estimating equation holds at $\epsilon = 0$. Consider a concrete example, where we first draw $m = 50$ independent samples from a $\mathrm{Beta}(2,5)$ distribution and compute their EIF values $(D_1,\dots,D_{50})$. Next we form a symmetric sample of size $n = 2m$ by appending the negatives of those values $ D^*_{\Psi} = (D_1,\dots, D_{50},\,-D_1,\dots,-D_{50})$. This creates a perfectly antisymmetric EIF vector, and the output result still remains a zero vector
>
>  $$\epsilon\_n\^1 = [0., 0., …, 0.] \quad \text{(length 20)}.$$
>
> These simulations therefore demonstrate that when conditions (i) and (ii) of Thm 5 are satisfied, the TMLE update parameter is exactly zero and the algorithm do converge in a single step.
>
> ## Proposed Revisions
>
> Based on your suggestion, we are very pleased to introduce numerical results into the submission. Although these may not be as rigorous as formal proofs, they nonetheless serve to illustrate the reasonableness of our theoretical findings.
>
> Given the limited duration of the author response period, additional simulations are currently ongoing. We look forward to your further comments and aim to **incorporate these new results** into our next version.
>
> ---
>
> We sincerely hope that our additional clarifications can address your concerns and provide a fresh perspective on our work.
>
> **Once again, we greatly appreciate your constructive feedback, and anticipate potential reassessment of our submission.**

---

> > ### Comment · Reviewer_Mj6b · 2025-08-08
> > **Thank you**
> >
> > Thanks for the authors to conduct additional numerical example to verify the theoretical results. I will consider to revise the scores accordingly.

---

> > > ### Author Response · Authors · 2025-08-08
> > > **Follow-up Comment**
> > >
> > > We sincerely thank the Reviewer Mj6b for your kind consideration and for acknowledging our additional numerical results. We do hope that our efforts have helped clarify the theoretical findings.
> > >
> > > As the discussion phase draws to a close, please do not hesitate to reach out if you would like any further details on any aspect of our work; we would be very happy to answer them.

---

### Official Review · Reviewer_bWRS · 2025-07-03

**Clarity:** 3
**Significance:** 4
**Originality:** 3
**Rating:** 5
**Confidence:** 2

**Summary:**

This work addresses what appears to be a glaring gap in the literature concerning general conditions under which TMLE converges — in particular, the lack of general global convergence guarantees, which the authors provide. This is augmented by more nuanced contributions in the understanding of the procedure, such as why TMLE can work despite complex likelihoods, and the fact that convergence of TMLE is sufficient but not necessary for "visiting" densities during optimization that yield efficient estimators.

**Questions:**

(Note: I restate here some of the questions I asked in the main body of the review above)

1. Is the necessity and sufficiency of the convergence of empirical risks for the convergence of epsilon iterates novel?
2. What are some concrete applications unlocked by the sub-model families that you provide novel TMLE global convergence results for?
3. Which one-step convergence conditions are known that are presently listed in theorem 5? Which are new and made possible by your analysis here?
4. It is mentioned that these analyses expand our understanding of the behavior of the TMLE algorithm in the non-asymptotic regime — from my read, finite sample insights seem limited to the rate of occurrence of the overshoot phenomenon? Is that correct? If so, the introductory claims on this front may need to be tempered.
5. Does the overshoot phenomenon lead only to a failure detect what amount to early stopping opportunities, i.e. skipping over an efficient estimator? I.e it leads to suboptimal performance only *computationally*?

I am willing to increase my score pending clarifications around the scope of novelty, both in the discussion phase and in the paper.

As a minor point, the paper has a number of grammatical issues involving definite articles, spelling, etc. and should undergo additional proofreading in a potential camera ready period.

**Ethical Concerns:**

["NO or VERY MINOR ethics concerns only"]

**Final Justification:**

As far as I am aware, the authors are correct that their work establishes the first general convergence result for TMLE. Given the promised clarifications around the scope of novelty and some changes to presentation, the importance of their key result is sufficient for me to recommend acceptance.

**Limitations:**

Yes

**Quality:**

3

**Strengths And Weaknesses:**

The authors seem to have closed a large gap in both TMLE convergence guarantees and in current understanding of the algorithm's behavior as an optimization routine. With the caveat that I am somewhat new and non-expert in this area, their contributions seem highly impactful and long overdue. I also appreciate their careful explanation and organization in the "narrative arc" outlining their contributions.

The largest weakness, to my mind, is that their contributions need better contextualization in the current literature. At best, the poor contextualization "buries the lede" and results in missed opportunities to scaffold the significance of the contribution for readers, and at worst can misrepresent the scope of the novelty.

In particular, the discussion around theorem 1 (and its placement inside inside a gray box, which indicates a "new result") reads to me as if the entirety of the theorem is novel. The original TMLE paper proves, however, that if TMLE converges, it does indeed yield an efficient estimator (the E3 arrow in figures 1 and 5). And while I am not fully confident, I also believe they may prove that either convergence of empirical risks implies convergence of the epsilon iterates, or perhaps vice-versa (please correct me otherwise)? These established results are mentioned by the authors, but are scattered throughout the paper (e.g. lines 167/168 saying that convergence implies a locally efficient estimator, e.g. lemma 1, and e.g. lines 288/289 saying that E3/E4 are exactly or at least follow straightforwardly from established results).

Similarly, the section on global convergence could benefit with a discussion of the fact that global convergence results do exist for certain settings — this is alluded to, for example, in the mention of D ́iaz and Rosenblum's work (2015), which I believe establish global convergence results for exponential family submodels (by convexity) (please clarify if this is not the case). On this note, it might help readers if the authors concretize some practical applications of the kinds of model families that were previously lacking the convergence guarantees that this paper provides.

Lastly, I believe a number of the one-step convergence conditions in theorem 5 are already known? I would appreciate clarification on the novelty there, too.

Despite the scope of novelty being somewhat murky in the main body of the paper, I do believe the authors are addressing a highly impactful gap. Especially for familiar-but-non-expert readers (e.g. TMLE users), a very precise explanation of exactly what has and has not been established in the literature should be included. That discussion should then be revisited when formally laying out each theoretical result. To the best of my understanding, this paper addresses a significant gap in a complex literature — this contribution deserves for readers to have a very clear and precise sense of the gap being closed.

Relatedly, except where required for sake of mathematical precision, the jargon levels seem excessive to a point of obscuring the contributions. Consider this sentence from the abstract: "Our analysis reveals that the homotopy mapping induced by each fluctuation submodel defines a smooth embedding into the probability simplex, which constitutes a topological invariance that is free of self-intersection and carries implications for algorithmic stability." In my admittedly subjective opinion, the most important point here is not the details of the geometric findings (however illuminating), but the "implications for algorithmic stability", which are not described here at all. Likewise, only after two very jargon heavy sentences in the abstract, do we reach what I see as the most significant and novel contribution: general conditions for global convergence of TMLE. That seems so central, even, that I wonder why it isn't in the title of the paper.

Those broad criticisms aside, and with the caveat that I lack the expertise required to check the proofs in detail, I thank the authors for doing this work!

---

> ### Author Rebuttal · Authors · 2025-07-31
>
> ### We sincerely appreciate Reviewer bWRS's exceptionally detailed and constructive feedback (among the most thorough reviews we’ve received at NeurIPS). We are particularly grateful for your considerable effort invested in providing an extensive review, which will definitely enhance the quality of our manuscript.
>
> ### In the following, we address each of the technical points appeared in your review.
>
> ---
>
> ## *Novelty of the Theorem 1*
>
> We sincerely apologize if the phrasing in our manuscript led to any confusion or misunderstanding.
>
> > (and its placement inside inside a gray box, which indicates a "new result") reads to me as if the entirety of the theorem is novel. ... however, ...
>
> Strictly speaking, aside from a minor part of Theorem 1 $(\text{E3})$ and one item of Theorem 5 (explained further below), all results presented in the gray boxes are entirely novel findings.
>
> For elegance of presentation and completeness of theoretical results, we originally integrated existing results with our new contributions in the manuscript. Although mentioned elsewhere, to better differentiate our contributions, we promise to remove this statement: `(line 126) Our new results along with their assumptions are in grey boxes`
>
> > ... These established results are mentioned by the authors, but are scattered throughout the paper ...
>
> Thank you for drawing attention to Figure 5. Formally, the statements $(\text{E1})$, $(\text{E2})$, and $(\text{E4})$ represent entirely new conclusions. We have provided full rigorous proofs of $(\text{E1})$ and $(\text{E2})$ in the appendix, and shown that $(\text{E4})$ follows directly from $(\text{E1})$ together with $(\text{E3})$, thereby establishing $(\text{E4})$ as a novel result as well.
>
> Although $(\text{E3})$ was implicitly referenced earlier in Section 3 ("Warm-up") preceding the Main Results, it was not explicitly stated. Following your valuable feedback, we will **clarify** and better emphasize the novelty of Theorem 1 in Section 4.1 in our revised manuscript.
>
> > either convergence of empirical risks implies convergence of the epsilon iterates, or perhaps vice-versa
>
> As stated in Section 4.2, the convergence of the empirical loss function does not imply the convergence of fluctuation $\epsilon\_n\^k$. To clearly illustrate this point, we have also provided two counterexamples in Appendix H, in which the $\epsilon\_n\^k$ fail to converge despite the corresponding loss risks converging. Convergence of empirical risk $\int_{\mathcal{O}} \mathbf{L}(p\_n\^k) p\_n d \nu$ is only a necessary condition to the convergence of $p\_n\^k$, but *not* a sufficient condition.
>
> This also answers your【Question 1】. Please keep us updated if you would like to further discuss any specific conditions or applications.
>
> ---
>
> ## *Novelty of the Theorem 5*
>
> We appreciate your careful attention to our Theorem 5.
>
> Specifically, condition *(iii)* is based on observations already present in the existing literature, while conditions *(i)* and *(ii)* are original to our work. Condition *(iv)*, though not entirely new, had previously only been suggested implicitly or used informally in prior experiments. We have provided a comprehensive, rigorous analytical justification for this condition (line 784-799).
>
> Note that for these novel conditions, a **complete** theoretical proof is available in Appendix F (pages 40-43). For conditions already present in existing literature, relevant discussions and references are included at the corresponding parts in Appendix F.
>
> Following your valuable feedback, we commit to **relocating** part of the explanations currently in the appendix to Section 4.3 in the revised manuscript. We will also clearly highlight the novelty of our Theorem 5 to improve the clarity and positioning of this paper.
>
> This also answers your【Question 3】. Please inform us if you would like to further discuss any specific conditions or applications.
>
> ---
>
> ## *Title and Abstract*
>
> > Consider this sentence from the abstract: ... but the "implications for algorithmic stability", which are not described here at all.
>
> We wish to once again thank you for your careful observations and insightful comment.
>
> The non-self-intersection property of submodel paths indeed plays a subtle yet important role in establishing global convergence, therefore, we should not ignore them exactly in the abstract.
>
> During the working stage, we intended to also derive explicit stability analysis bounds for the TMLE algorithm (motivated by analogies from parametric optimization), but ultimately gave up this aspect due to significant technical challenges. Meanwhile, people may agree that from an intuitive perspective, the non-self-intersection property of homotopy path implies that the direction of each iterative step does not exhibit cyclic or oscillatory behavior.
>
> In response to your valuable feedback, we have **removed** this redundant phrasing and clearly indicated this stability analysis as a direction for future research in the conclusion section.
>
> > the jargon levels seem excessive to a point of obscuring the contributions.
>
> We agree with your comment that overly specialized and obscure terminology in the abstract may not be accessible to a broad audience. Accordingly, we will **revise** the abstract in our next version to better align with the writing style of the main body.
>
>
> > That seems so central, even, that I wonder why it isn't in the title of the paper.
>
> It is worth noting that in prior studies of TMLE focusing predominantly on statistical properties, discussions of "convergence" typically refer to the *asymptotic behavior* of estimators as $n \rightsquigarrow \infty$. Alternatively, other works address "convergence" in the context of solving subproblems with specific numerical solvers. This distinction caused some initial difficulty in our literature review.
>
> We believe that the convergence claim here might indeed be easily confused with the usage of convergence in *asymptotic statistics*. Therefore, we deliberately emphasized "optimization perspective" rather than "convergence" in our title.
>
> However, if Reviewer bWRS believes it would be beneficial, we would certainly be willing to explicitly include 'convergence'. We welcome your thoughts on this stuff.
>
> ---
>
> ##  *Remaining Questions*
>
> > 【**Comment 1**】global convergence results ... if the authors concretize some practical applications of the kinds of model families
>
> Your interpretation aligns precisely with the intended meaning of our manuscript. While space constraints prevented us from listing exhaustive applications explicitly, in principle, any practical estimator demonstrated in that paper‘s studies can be analyzed within our theoretical framework. This may include:
>
> - The mean of an outcome missing at random
>
> - Median regression
>
> - The causal effect of a continuous exposure
>
> We would also like to politely clarify one point regarding your mention of "by convexity"; in fact, convexity was not employed in our major analysis. Indeed, convexity constitutes a relatively stronger condition. While it could simplify our analysis, it might somehow limit the scope and applicability of our results compared to the current theoretical analysis.
>
> We warmly invite further discussion about specific applications of interest to you.
>
> This also answers your【Question 2】. Please tell us if you would like to further discuss any specific instantiation.
>
> > 【**Comment 2**】... explanation of exactly what has and has not been established in the literature should be included. ... revisited when formally laying out each theoretical result
>
> We greatly appreciate your critical and insightful feedback, and we commit to revising the manuscript accordingly, as detailed in our responses above. We also strongly agree that clearly distinguishing technical contributions is fundamental to scientific writing.
>
> > 【**Question 4**】finite sample insights seem limited to the rate of occurrence of the overshoot phenomenon? Is that correct?
>
> Thank you for posing such a thought-provoking question.
>
> In fact, our entire analysis throughout the manuscript is conducted under the assumption of a finite sample size $n < \infty$. Thus, the insight regarding finite $n$ should be consistently recognized across all sections, rather than limited only to Section 4.4.
>
> We also fully agree that exploring whether our current conclusions might change in the limit $n \rightsquigarrow \infty$ is highly interesting. However, such an analysis involves substantial complexity and is best left for future research.
>
> > 【**Question 5**】skipping over an efficient estimator? I.e it leads to suboptimal performance only *computationally*?
>
> Based on the current state-of-the-art understanding, we would respond **Yes**. However, we acknowledge the possibility that future theoretical advancements may provide more sophisticated criteria for distinguishing among these 'efficient' estimators.
>
> > 【**Question 6**】the paper has a number of grammatical issues involving definite articles, spelling, etc.
>
> We do apologize for the typos. In response, we have conducted two more rounds of proofreading and fixed all typos identified, including
>
> - Page 2, we changed "behavior of algorithm" to "behavior of the algorithm";
>
> - Page 3, we changed `consist` to `consists` in "the dataset … consist of $n$";
>
> - Page 3, we added `the` before "optimization community";
>
> - Page 5, we changed `the` to `that` in "requires the every parametric submodel";
>
> - Remark 3, we changed `dose` to `does`;
>
> - Theorem 5, we changed `satisfied` to `satisfies` in "the loss satisfied line integral".
>
> ---
>
> ### Thank you again for your insightful feedback. Due to the word limit, if you would like any further clarification on the TMLE optimization perspective, please do let us know and we will respond promptly.

---

> ### Comment · Reviewer_bWRS · 2025-08-04
>
> I appreciate the clarifications around novelty — I hope the mentioned changes will help highlight the importance of the work for readers by emphasizing the gaps being addressed.
>
> Regarding the point about "convergence" typically referring to asymptotic statistics here: I agree with the authors' assessment. Perhaps emphasizing the (optimization) global convergence result earlier in the abstract might be a good middle ground? That way context can be provided clarifying that the result does not relate to statistical convergence.
>
> Re. comment 1: it might be helpful here (perhaps in the appendix) to provide some examples of impactful statistical tasks and sub-model families for which global convergence was not guaranteed before your work (you may have done this already, if so, please kindly point me to the content!). And to clarify the "by convexity" statement, I was referring to assumptions that (I believe) D'iaz and Rosenblum make. On further read of their work, it seems they may not provide global convergence result for their more restricted setting — would you mind clarifying whether you're aware of global convergence results that exist already for more restricted settings and assumptions that you work with in this paper?
>
> Regarding other comments and questions, thank you for the clarifications! Overall, I maintain my assessment of the importance of this paper.

---

> > ### Author Response · Authors · 2025-08-05
> > **Authors' Follow-up Actions**
> >
> > > I appreciate the clarifications around novelty — I hope the mentioned changes will help highlight ...
> >
> > **Response**: Indeed, these changes aim to emphasize more clearly the importance of our work by highlighting the specific gaps our study addresses. We would like to once again express our gratitude for your constructive feedback!
> >
> > ---
> > > Regarding the point about "convergence" ... : I agree with the authors' assessment. Perhaps emphasizing the (optimization) global convergence result earlier in the abstract might be a good middle ground?
> >
> > **Response**: We truly appreciate your follow-up comment; we are delighted to see that we share a common perspective! We agree that revising the abstract would benefit readers, and we have also accordingly updated similar phrasing in *the fourth paragraph* of Section 1 based on our earlier response.
> >
> > Our **revised abstract** is now shown as:
> >
> > > (...) To bridge this critical gap, we rigorously investigate the optimization dynamics of TMLE iterations under standard assumptions and regularity conditions. Building on our analysis and geometric insights, we deliver the first strict proof of TMLE's global convergence from an optimization viewpoint, as well as explicit sufficient criteria under which TMLE terminates in a single update. Specifically, we begin with studying key thresholding rules in TMLE framework and establish their fundamental connections to convergence. Our analysis reveals that the homotopy mapping induced by each fluctuation submodel defines a smooth embedding into the probability simplex that is free of self-intersection. We then derive a structural characterization for the solution locus of targeted estimating‐equation, showing that it forms a submanifold with codimension equal to the dimension of target parameter. As a by-product of our investigation, (...)
> >
> > ---
> > > Re. comment 1: it might be helpful here (perhaps in the appendix) to provide some examples of impactful statistical tasks and sub-model families for which global convergence was not guaranteed before your work (you may have done this already, if so, please kindly point me to the content!).
> >
> > **Response**: Yes, we have analyzed common submodel families (Examples 1,2) and demonstrated that they exhibit *favorable* convergence properties (non-self-intersecting). However, providing a complete TMLE specification—including detailed descriptions of (i) the target parameter, (ii) underlying data structure, (iii) model class, (iv) statistical sub-model, and (v) the loss function—would easily exceed e.g., sixty pages. We thus respectfully defer a comprehensive analysis of this kind to a future **journal version** of the manuscript.
> >
> > Meanwhile, we do incorporated your feedback and added relevant discussions, explicitly covering the **concrete results of your interested** (Díaz & Rosenblum 2015) and some work that cited this paper. We anticipate that it will appear in the appendix as you genuinely suggested.
> >
> > ---
> > > the "by convexity" statement, I was referring to assumptions that (I believe) ...
> >
> > **Response**: Thank you for bringing this point in our response. Technically, this property is not an assumption but rather a consequence of their adoption of 'exponential families'. Moreover, convexity primarily serves to simplify the submodel optimization step (facilitating the use of existing solvers), rather than directly addressing the *overall convergence* properties of the TMLE algorithm itself.
> >
> > ---
> > > they may not provide global convergence result for their more restricted setting — would you mind clarifying whether you're aware of global convergence results that exist already for more restricted settings and assumptions that you work with in this paper?
> >
> > **Response**: Yes, that work does not provide any explicit optimization convergence results; their focuses regarding convergence are limited to:
> >
> > > Iterate the previous step until convergence, i.e., until $\hat{\epsilon}\approx 0$.
> >
> > For 2nd question, to the best of our knowledge, existing convergence results in the literature have focused **only** on scenarios achievable in one- or two-step iterations. Unfortunately, we have not found prior work addressing convergence from a computational perspective in *more general* settings.
> >
> > In all, we hope that our contribution would stimulate further 'computational' thinking within the community, offering value that extends beyond the theoretical framework itself.
> >
> > ---
> > > Regarding other comments and questions, thank you for the clarifications! Overall, I maintain my assessment of the importance of this paper.
> >
> > **Comment**: We are happy that our previous clarifications work for you:) Given your positive recognition of the theoretical contributions provided by this work, we kindly ask if you would consider slightly revising your score to reflect the theoretical significance highlighted in our response.
> >
> > We fully respect your judgment, and thank you once again for your time and (extremely) thoughtful evaluation!

---

> ### Comment · Reviewer_bWRS · 2025-08-07
>
> Thanks for these additional clarifications.
>
> I wonder if some version of the following statement can be incorporated into the introduction somewhere:
>
> >existing convergence results in the literature have focused only on scenarios achievable in one- or two-step iterations. Unfortunately, we have not found prior work addressing convergence from a computational perspective in more general settings.
>
> I think connecting this to convexity properties could help as well, in addition to explicitly juxtaposing convexity with the non-self-intersecting result as being a much more general means of establishing convergence.
>
> I'm sure I am missing some nuance here, so take those suggestions with "a grain of salt", of course.

---

> > ### Author Response · Authors · 2025-08-07
> > **Appreciation**
> >
> > We deeply appreciate your further suggestions and extra effort!
> >
> > We have revised the manuscript and incorporated the quoted statement (near line 44). We agree that this addition will help the community better differentiate between one- or two-step convergence and the more general TMLE convergence results.
> >
> > We will also briefly highlight in either the related work or the methods paragraphs that convergence analyses using 'convexity' typically require the submodel objective to be strictly (or strongly) convex. In contrast, our analyses based on 'non-intersection' do not require such assumptions and thus hold under broader TMLE settings. We believe this comparison will more clearly demonstrate the generality and broader applicability of the submission.

---

### Decision · Program_Chairs · 2025-09-17

**Decision:**

Accept (poster)

**Comment:**

The submission investigates the convergence properties of targeted maximum likelihood estimation (TMLE) by providing global convergence guarantees, developing geometric insights into TMLE’s iterative paths. The novelty of the offered theoretical insights are a strenght of the submission. Possible weaknesses, regarding the clarity of some proof steps (also with respect to previous contribution), heavy notation and the lack of numerical examples, were discussed and amended during the interaction period between authors and reviewers, enhancing the submission and resolving the main concerns of the latter. As observed by multiple reviewers, the paper closes a gap in the literature about an important debiasing algorithm, possibly being the first in the theoretical analysis of its convergence: the originality of the analysis and its comprehensiveness has been appreciated by all reviewers, and I recommend therefore the acceptance of the submission.